# One-Step Solution Plasma-Mediated Preparation of Se Nanoplarticles and Evaluating Their Acute Oral Toxicity in Mice

Tri Thien Vu [1], Dung Thi Nguyen [2], Tran Hung Nguyen [1,*], Huu Thanh Le [1], Dinh Duc Nguyen [3,*] and Duong Duc La [1,*]

1   Institute of Chemistry and Materials, Nghia Do, Cau Giay, Hanoi 100000, Vietnam
2   School of Chemical Engineering, Hanoi University of Science and Technology, 1 Dai Co Viet, Hanoi 11615, Vietnam
3   Department of Environmental Energy Engineering, Kyonggi University, Suwon 16227, Korea
*   Correspondence: nguyentranhung28@gmail.com (T.H.N.); nguyensyduc@gmail.com (D.D.N.); duc.duong.la@gmail.com (D.D.L.)

**Abstract:** Selenium element is considered as one of the most important micronutrients for many biological systems. It has been well demonstrated that Se nanoparticles (Se NPs) express greater bioavailability, biocompatability, and less toxicity than that of Se in ion form. In this work, the Se NPs were facilely fabricated by a one-step plasma process in the ethanol–water solution mixture. The as-prepared Se NPs were well characterized by scanning electron microscopy (SEM), X-ray diffraction (XRD), UV-vis spectroscopy, Raman spectroscopy, and energy dispersive X-ray spectroscopy (EDX). The prepared Se NPs were a light red color with a spherical shape and particle size in the range of 100–200 nm. The average diameter of the Se NPs calculated from the ImageJ software and TEM image was approximately 154 nm. The EDX results showed that the Se NPs prepared by the plasma process in the solution were highly pure and stable. The acute oral toxicity of the obtained Se NPs toward mice was also studied, which revealed that the Se NPs were safe for the human body. The mechanism for the formation of Se NPs from the Se ions under the solution plasma condition was also studied and discussed.

**Keywords:** selenium nanoparticles; solution plasma; green synthesis; nanoparticles characterization; acute toxicity

## 1. Introduction

Organic compounds or inorganic ions have been extensively employed as supplement nutrients or applied in many biological processes [1,2]. Selenium (Se) element is of great significance in many fields such as physics, chemistry, and biology. In nature, the Se element commonly exists in two forms of inorganic (selenium and selennate) and organic (selenomethiomine and selenocysteine). They could be found in both crystalline (single or triangle crystalline) and amorphous nature [3]. Selenium is one of the most important micronutrients for many biological systems including, but not limited to, antioxidant effects, anticancer, and antiviruses activities [4]. A deficiency of the Se element could lead to serious diseases such as cancers, heart desease, immune disoder, or immune inhibiting. Yet, the supplement of the Se nutrient could increase or recover immune functions [5]. Thus, it is important to monitor the Se content in the body to avoid the development of undesired diseases. In comparison to Se in ion form, Se nanoparticles (Se NPs) are considered as expressing greater bioavailability, biocompatability, and less toxicity [6–9]. The biological properties of the Se NPs mostly depends on the particle size and the morphology; a smaller particle size enables a higher biological activity of the Se NPs. Se NPs could be synthesized in mophorlogies of nanorods, nanowire, nanospheres, and nanotubes. Many approaches

have been successfully employed to obtain Se NPs including, but not limited to, chemical synthesis [10,11], refluxing [11], microwave [12], hydrothermal [12,13], gamma radiation, laser sputtering [14], and physical methods [15,16].

Plasma has been well known as the fourth state of material at which material can exist in the form of electronic-conveyed particles such as ions and electrons. Based on the thermodynamic properties, plasma is divided into two basic categories of thermal plasma (heat enquilibrium plasma) and non-thermal plasma (heat non-enquilibrium plasma) [17]. Plasma could be also categorized into plasma chemistry [18] and plasma physics [19] based on the physical properties. Plasma technology has been extensively utilized in many fields of application including, but not limited to, medical, environmental treatment, synthesis of materials, and electronics [20,21]. Among these, the use of plasma for the synthesis of materials has attracted great attention from scientists because of their quick syntheis and environmental friendliness. Electrochemical plasma is generated from two electrodes when a voltage of a few to several dozen kV is applied to the electrodes. Under this strong electric field, subtances such as $H_2O$, oxygen, and other gasses at the surrounding electrode are ionized to form plasma [22]. Electrochemical plasma formed in the solution consists of two separate regions: the plasma region around the electrodes and the liquid phase near the plasma region called the contact surface region. At the plasma region, $H_2O$ is vaporized and dissociated to form $H_2$ and $O_2$. The redicals such as H•, O•, and OH• are also generated and diffused into the solution [23]. These radicals with highly reactive activities could reduce metal ions to form metals in the form of nanostructures.

Many metals' metal oxide nanopaticles have been successfully prepared using the cold plasma in solution [24–32]. For example, the one-step solution plasma process was employed to synthesize black $TiO_2$ nanoparticles with superior solar-thermal water evaporation efficiency [33]. Nanoscale Si particles were also synthesized using the solution plasma process at ambient conditions [34]. The prepared Si NPs with an average diameter of lower than 10 nm were used to fabricate the anode for the Li-ion battery, which revealed a high reversible capacity of 537 mAh/g after 30 discharge/charge cycles at a current rate of 0.5 A/g. In another study, Yoshida et al. successfully farbicated silver nanoparticles in an aqueous solution of ammonia by a solution plasma process and studied their photocatalytic performance for carbon dioxide reduction with water [35]. The solution plasma process is also considered as an effective technique to control the size and morphologies of nanoparticles by adjusting the plasma conditions and reaction media. Even with the many advantages of using the solution plasma process for the preparation of nanoparticles, the use of solution plasma for the synthesis of the Se NPs has been rarely investigated. The biological properties of the Se NPs prepared by the solution plasma process have not been studied in the literature.

Herein, we report a facilely one-step preparation of the Se NPs using the solution plasma process at ambient conditions. The prepared Se NPs are thoroughly characterized using scanning electron microscopy, transmittance electron microscopy, X-ray diffraction, Raman spectroscopy, and energy dispersive X-ray spectroscopy. The acute oral toxicity and repeated 7-day dose oral toxicity of the Se NPs in mice are also investigated in detail.

## 2. Materials and Methods

### 2.1. Materials

Selenium dioxide ($SeO_2$, 99.99%) was received from Sigma-Aldrich. Ethanol (99.7%) and NaCl (99.9%) were obtained from Xilong chemicals (China). The $H_2SeO_3$ solution was prepared by dilution of $SeO_2$ in double distilled water with a concentration of 3 mM. Double distilled water was employed in all experimental processes. All chemicals were used as received without any additional purification.

### 2.2. Solution Plasma Setup and Synthesis of Se NPs

A schematic experimental setup for synthesizing the selenium nanoparticle without employment of any chemical agents is shown in Figure 1. The electrode was made of

tungsten with a 99.9% purity and 3 mm in diameter. The distance between electrodes was in range of 20 to 30 mm. The supply power of 1 kW, output voltage ranging from 0 to 10 kV, frequency of 100 Hz–30 kHz, and pulse width of 0–10 μs were used as the plasma source and provided by PEKURIS company (Japan). The discharge inside the solution was steadily conducted for 2 h in the reactor with a magnetic stirring speed of 250 rpm. A water cooling system was employed to control the reaction temperature and the oscilloscope was used to monitor the stability of the plasma discharge. Plasma at the electrode ends formed in the solution and reacted with the water and ethanol to form free radicals. These radicals participated in the reduction in Se ions to the metallic Se in nanostructures. The color of the reacting solution gradually changed from light yellow to a red brick color during the plasma time, confirming the formation of Se NPs from Se ions.

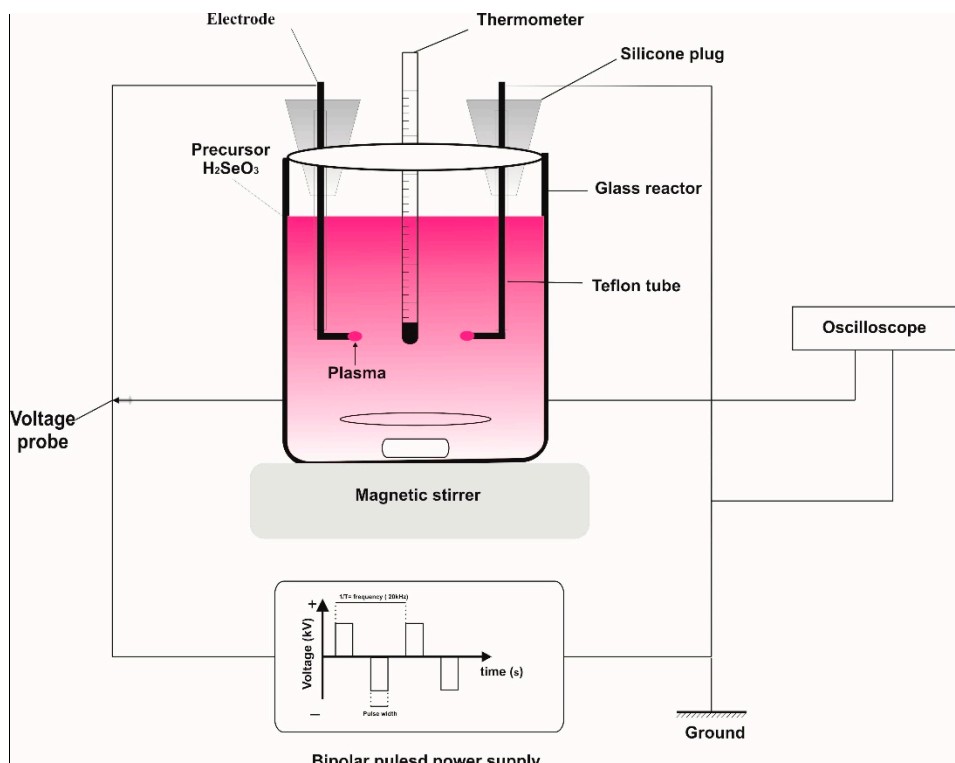

**Figure 1.** The diagram of the solution plasma process for the synthesis of selenium nanoparticles.

### 2.3. Acute Toxicity

Forty-eight healthy BALB/c white mice, with a weight of about 22–26 g, regardless of breed, were raised in the animal house under standard conditions of temperature and light, then they were divided into eight lots (six rats/lot) and starved for 16 h before drinking the study sample. Se NPs dispersed in 0.9% pathogen-free NaCl were administered orally at different doses. The animals were observed for survival and clinical signs of toxicity on the dosing day and then daily for 14 days. The cumulative mortality rate within 72 h of treatment was used to calculate the mean fatal dose ($LD_{50}$).

### 2.4. Characterization

The morphology of the Se NPs was observed *via* field emission scanning electron microscopy (SEM-EDX, Hitachi S-4600). The detailed structure was analyzed *via* transmission electron microscopy (TEM, JEM-2100F, JEOL) and high-resolution transmission electron microscopy (TEM, JEOL JEM 1010). The elemental analysis of the prepared Se NPs was studied using energy-dispersive X-ray spectroscopy (Hitachi S-4600). The crystal structure of Se NPs was obtained on an X-ray diffractometer (XRD, X'Pert PRO PANalytical) with Cu Kα ($\lambda$ = 1.5418 Å) radiation operating at 45 kV and 200 mA. The XRD patterns were

acquired in the range of 20–70° with a step size of 0.01° and a scan speed of 1 min$^{-1}$. The Raman analysis was performed on the DXR3 Thermal scientific instrument. UV-vis spectrophotometer (Jasco V730) was utilized to investigate the surface plasmon resonance of the prepared Se NPs.

## 3. Results and Discussion

### 3.1. UV-Vis Spectrum of SeNPs

The Se NPs were prepared by the solution plasma process in the ethanol–water solution mixture. It has been demonstrated that by adding ethanol in the reaction media, the plasma-mediated synthesis of nanoparticles could significantly accelerate [36]. The presence of ethanol in the reaction solution also improves the hydrogen bond's stability in the network between water molecules; as a result, changing the surface properties of the reaction media. When a high voltage was applied to the electrodes, a plasma was formed in the mixture solution between two electrodes. Electrons generated from the plasma react with ethanol and water to form readicals of H•, O•, and OH•. These radicals participate in the reduction in Se$^{4+}$ ions to the metallic Se NPs. The formation of Se NPs was primarily confirmed by the UV-vis spectroscopy, as illustrated in Figure 2. It could be visually observed that the colour of the solution changed reapidly when the plasma appeared between the electrodes. The colour of the reaction solution was light yellow before applying the voltage. After the appearance of plasma in the solution, the mixture gradually changed colour to the red and this change was completed after 60 min of plasma-induced reaction time. The surface plasmon vibration of the excitation state of Se NPs is responsible for the appearance of the red colour in the solution [37]. As can be seen in Figure 2, the Se$^{4+}$ ions exhibit no surface plasmon vibration effect with no absorption peak observed. However, the sample with the plasma irradiated for 60 min shows a sharp peak at the wavelength of 301 nm, which demontrates the formation of Se NPs with the surface plasmon vibration effect [38]. This result confirms the successful formation of Se NPs from Se$^{4+}$ ions under the irradation of plasma in the ethanol–water solution mixture.

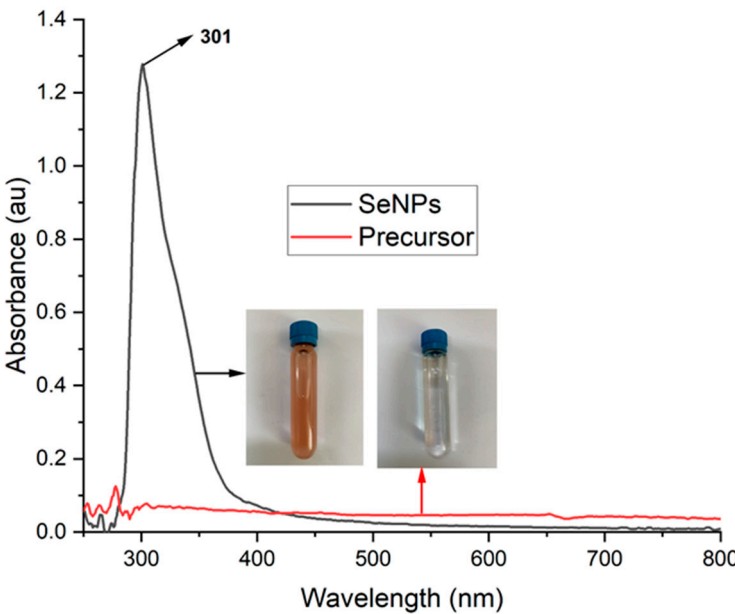

**Figure 2.** UV-vis spectra of Se ions (red line) and Se NPs (black line) formed by irradition of the plasma in the Se ions containing ethanol–water mixture after 60 min.

### 3.2. SEM and TEM Images of SeNPs

The formation of the Se NPs was further confirmed by the SEM and TEM images. Figure 3 shows the morphologies of the SE NPs prepared by the one-spot plasma process in the ethanol–water solution mixture. It can be seen from the SEM images (Figure 3a,b)

that the Se NPs are formed in the sphere shapes with the diameter of less than 200 nm. The high resoltuion of the TEM images (Figure 3c,d) clearly confims the spherical structure of the prepared Se NPs with a particle size ranging from 100 to 200 nm. The ImageJ software was employed to determine the average diameter of Se NPs from the TEM image (insert in Figure 3c). From the particle distribution diagram obtained by the ImageJ software, the average diameter of the Se NPs is calculated to be approximately 154 nm. The SEM and TEM images also reveal the well-distribution of the Se NPs with no aggregation, which demosntrates the high stability of the Se NPs prepared using the plasma process in the ethanol–water mixture.

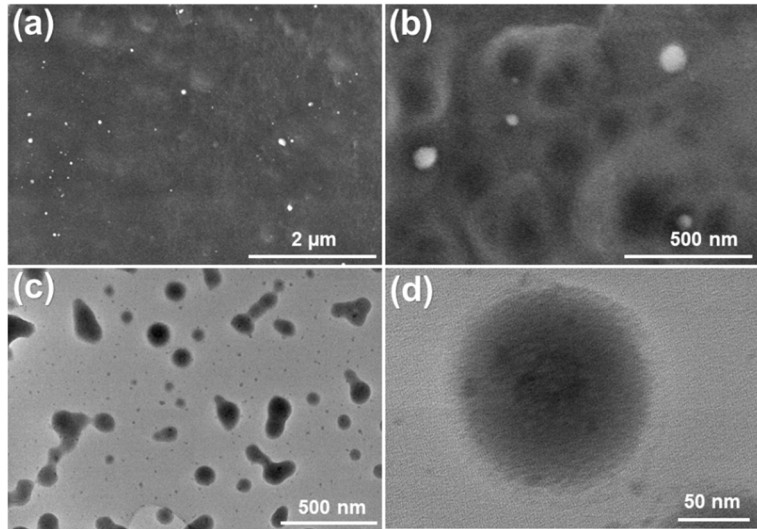

**Figure 3.** SEM images (**a**,**b**) and TEM images (**c**,**d**) of the Se NPs prepared by the plasma process in the solution mixture after 60 min of irradiation time.

### 3.3. EDX Spectrum, EDX Mapping, XRD Pattern and Raman Spectrum of the Se NPs

Illustrated in Figure 4 is the EDX spectrum and EDX mapping of the Se NPs prepared by the plasma process in the ethanol–water solution mixture. The EDX spectrum shows two characteristic peaks: a sharp peak with high intensity at 1.4 keV and a peak with low intensity at 11.2 keV, which are assigned to the characteristic peaks of the SeL$\alpha$ and SeK$\beta$ from metallic Se [39]. The apprearance of the characteristic peaks at near 2 keV is attributed to the Si substrate, which was used for the EDX measurement. No sign of the O observed in the EDX spectrum demonstrates that the Se NPs prepared by the solution plasma process in this study are highly stable. The high purity of the Se NPs was further confirmed by the EDX mapping. It can be clearly seen that only Se element is observed on the sample. This result demonstrates that the Se NPs obtained from the solution plasma process in the ethanol–mixture solution are highly stable without oxidation from oxygen in the solution. This might be due to the formation of the organic compounds during the plasma process in the ethanol–water solution; this newly formed organic compound could protect the Se NPs from the oxidation.

To further confirm the formation of the protective organic compounds on the surface of Se NPs after the plasma process, the FTIR spectrum of Se NPs were obtained and studied as shown in Figure 5. The FTIR spectrum shows broad absorption bands in the range of 3000–3500 cm$^{-1}$ that are assigned to the O-H stretch of alcohols and phenols, as well as the C-H stretch of alkynes [40–42]. The absorption band at 1645 cm$^{-1}$ is ascribed to the C=O bonds of the aldehyde and cetone groups, indicating the formation of organic compounds on the Se NPs' surface. The characteristic peaks at 1434 and 1078 cm$^{1}$ are attributed to the C-H and C-N stretches, respectively, of the phenol and fatty acid presented on the surface of the Se NPs. The C–X stretching in alkyl halides are also observed at the vibration bands of 795 and 608 cm$^{-1}$. These results confirm the presence of organic compounds in the

solution after the plasma process could form a protective layer on the surface of the Se NPs and prevent them from oxidation.

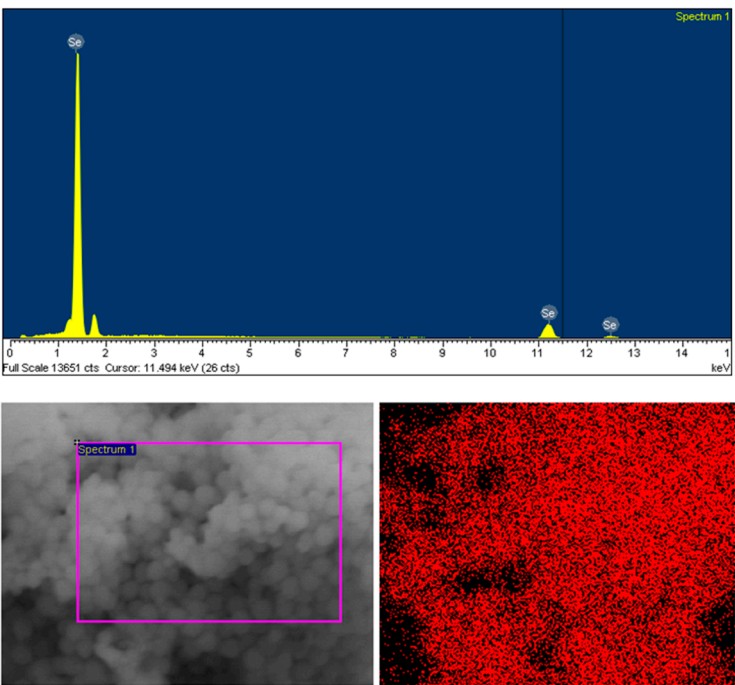

**Figure 4.** EDX spectrum and EDX mapping of the Se NPs prepared by the plasma process in the ethanol–water mixture after 60 min of reaction time.

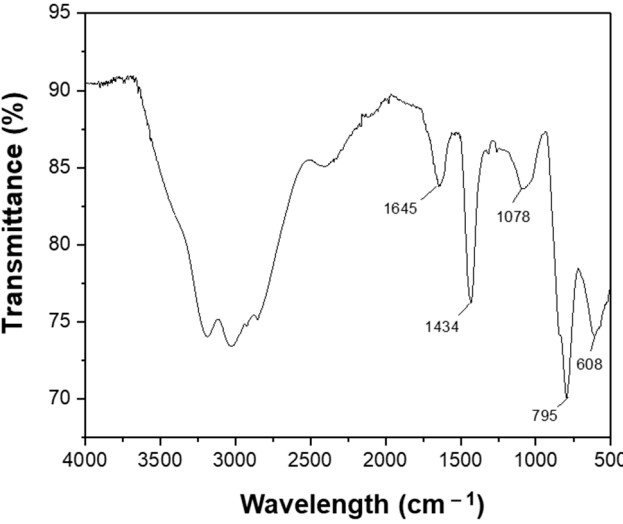

**Figure 5.** FTIR spectrum of the prepared Se NPs.

The crystallinity of the Se NPs prepared by the plasma process in the ethanol–water solution was investigated by the X-ray diffraction measurement and the results are shown in Figure 6a. The XRD pattern shows a sharp peak at around 30°, which is corresponded to the (101) plane of trigonal phase Se NPs with lattice constants of a = 4.366 A° and c = 4.956 A° (JCPDS file no. 06-362) [43,44]. The noisy diffraction peaks from the XRD pattern indicate that the Se NPs synthesized from the solution plasma process have low crystallinity compared to other methods [45]. The reduced crystallinity of the resultant Se NPs prepared by the solution plasma process was further confirmed by the Raman spectroscopy, as shown in Figure 6b. The appearance of a single, sharp peak at approximately

261 cm$^{-1}$ in the Raman spectrum of the Se NPs is assigned to the A1 symmetric stretching mode of amorphous Se [46,47].

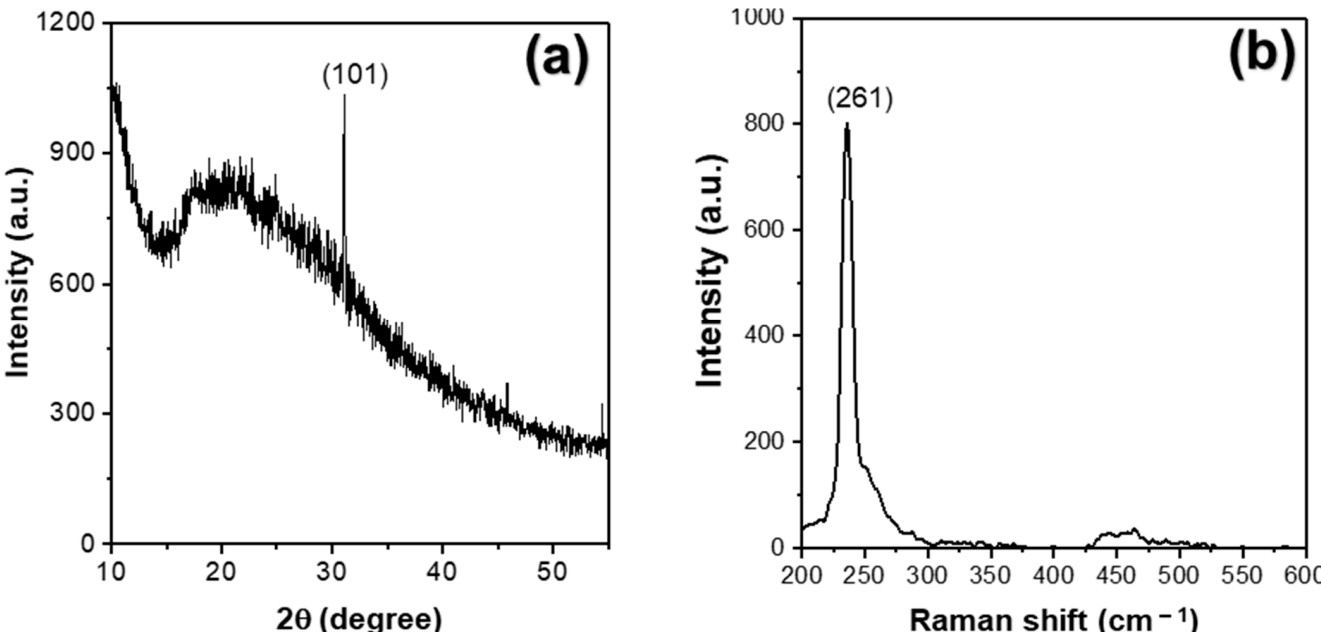

**Figure 6.** (**a**) XRD pattern and (**b**) Raman spectrum of the Se NPs prepared by the plasma process in the ethanol–water mixture after 60 min of reaction time.

### 3.4. Mechanism of SeNPs Formation by Plasma Solution

It has been well known that the plasma induced by applying a high voltage to the electrodes forms radicals such as H•, O•, and OH• when reacting ethanol with water to form radicals of H•, O•, and OH•. These radicals are responsible for the reduction in metal ions to the metallic nanoparticles. Based on this well-documented literature and the results above, a plausible mechanism for the formation of the Se NPs from Se ions is proposed, as shown in Figure 7. Under the plasma conditions, $H_2O$ and ethanol will generate free radicals such as H•, OH•, O• in the solution. These radicals defuse into the reaction media and participate in the reduction process of $Se^{4+}$ to metallic Se as illustrated in the following reaction:

$$H_2O + e = H\bullet + OH\bullet \tag{1}$$

$$SeO_3{}^{2-} + 6H\bullet = Se^0 + 3H_2O \tag{2}$$

$$CH_3CH_2OH + O\bullet = \bullet CH(CH_3)OH + HO \tag{3}$$

$$CH_3CH_2OH + OH\bullet = \bullet CH(CH_3)OH + H_2O \tag{4}$$

$$CH(CH_3)OH + SeO_3{}^{2-} = Se^0 + CH_3CHO + H_2O \tag{5}$$

The H• radical plays a major role in reducing Se ions to Se NPs, thus, in order to accelerate the Se NPs forming rate, the ethanol was added to speed up the generation of free radicals as well as to increase the concentration of radicals, therefore enhancing the formation of Se NPs from Se ions.

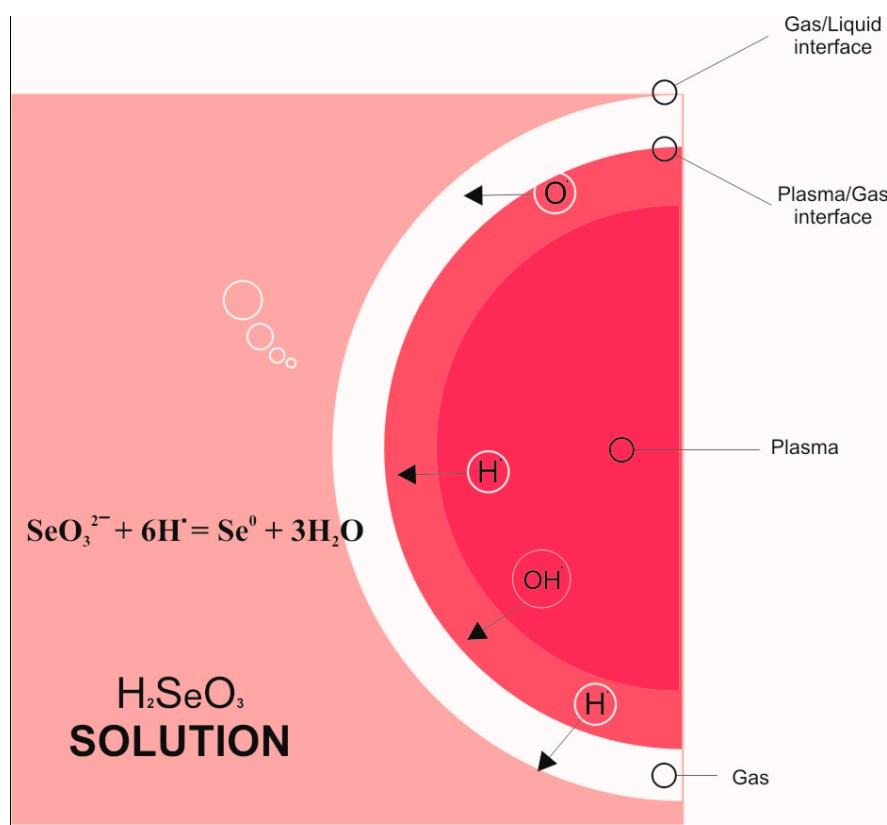

**Figure 7.** Mechanism for formation of Se NPs from Se ions under the solution plasma condition.

### 3.5. Assessment of Acute Toxicity in Rats

It has been well known that the acute toxicity and lethal concentration (LC50) are facile and acceptable protocol to investigate the toxicity and ecotoxicology of materials. The relationship between a specific effect of a sample and the dose at which the testing animal died is called the lethal concentration. The lethal concentration is determined from the 50% mortality of the experimental animals in a certain period of time, which is biologically and ecologically important to evaluate the toxicity of the materials used for the human and animals. In comparison to the selenium ions, such as selenomethionine, the Se NPs are of lower toxicity. In this work, the acute toxicity of Se NPs in terms of treatment and cumulative response at 72 h in mice was studied. The results show that at Se NPs' dose of less than 6 mg kg$^{-1}$, the mice move and consume the food normally, showing a good response to the light and sound, which is similar to the controls. However, when the dose increases to higher than 6 mg kg$^{-1}$, the mice start consuming the food and moving weakly, and 50% die within 72 h at Se NPs' dose of 9 mg kg$^{-1}$. However, this dose of Se NPs is nearly four folds higher than that of $SeO_2$ ($LD_{50}$ = 2.5 mg kg$^{-1}$). These results indicate that the chemical form, oxidative state, and reduced solubility of Se NPs significantly reduces the acute toxicity in comparison to the Se ions.

### 4. Conclusions

In summary, Se nanoparticles have been successfully synthesized by the plasma process in the ethanol–water solution mixture at the ambient conditions. The as-prepared Se NPs are in a spherical shape with a diameter in the range of 100–200 nm. The Se NPs prepared from the plasma process in the ethanol–water solution mixture was demonstrated to be low crystallinity in comparison to other synthesizing methods. The high purity and stability of the Se NPs were attributed to the protective layer from organic compounds formed during the plasma process in the ethanol–water mixture and coated on the surface of the newly-formed Se NPs. The acute oral toxicity of Se NPs was also investigated, which

showed that with the Se NPs' doses in the range of less than 6 mg kg$^{-1}$, no death or signs of toxicity in mice were observed. The LD$_{50}$ value of Se NPs was determined to be at dose of 9 mg kg$^{-1}$, which was four folds higher than that of SeO$_2$. During testing time, the mice consumed food and water normally, indicating the safety of Se NPs in the human body. With low particle size and high safety, the Se NPs are considered a promising supplement nutrient for body.

**Author Contributions:** Conceptualization, T.H.N., D.D.L. and D.D.N.; methodology, T.T.V., D.D.L. and D.T.N.; software H.T.L.; investigation, T.T.V., D.D.L., D.D.N. and D.T.N.; data curation—T.H.N. and H.T.L.; writing—original draft, T.T.V. and D.D.L.; writing—reviewing scientific contents and editing, T.H.N., D.D.L. and D.D.N. All authors have read and agreed to the published version of the manuscript.

**Funding:** This work was financially funded by the e-ASIA JRP (Project No. NĐT.74.e-ASIA/19).

**Institutional Review Board Statement:** Not applicable.

**Informed Consent Statement:** Not applicable.

**Data Availability Statement:** Not applicable.

**Acknowledgments:** The authors thank staffs at Institute of Chemistry and Materials for the support in characterizing and analysis. We also thank the Institute for all facilities we used to carry out the work.

**Conflicts of Interest:** The authors declare that no conflict of interest exist in the submission of this manuscript.

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
