# Peer review of "One-Step Solution Plasma-Mediated Preparation of Se Nanoplarticles and Evaluating Their Acute Oral Toxicity in Mice"

_sustainability, doi:10.3390/su141610294_

Round 1
Reviewer 1 Report
The work entitled “One-step solution plasma-mediated preparation of Se nanoplarticles and evaluating their acute oral toxicity in mice” presents interesting results on the fabrication of selenium nanoparticles using solution plasma process and studied their acute oral toxicity of the obtained Se NPs toward mice, which revealed that the Se NPs was safe for the human body. The manuscript is well-written with scientific sound and logic. There still exists some points and errors the authors should address before reconsideration for the publication: The novelty of the manuscript is too vague and not clear, it should be clearly stated in the introduction part; The mechanism for the enhanced synthesis of the Se nanoparticles by addition of the ethanol need to be clarified in the first part of discussion section.; More discussion on the mechanism of SeNPs formation by plasma solution should be added and proved; there are many English grammars and errors throughout the manuscript.
Author should include receive review articles (Chem.Soc.Rev. and Chem. Rev.) by Bhosale et al. to expand the work
The whole manuscript should be edited by a native English speaker or an editing service; Please carefully check the format and references style of the manuscript followed the author’s guide from the Sustainability journal.
Author Response
August 15, 2022
Sustainability
Re: Response on Manuscript ID 1852983:
Thank you for considering our manuscript for publication in Sustainability. We are very grateful for such a good news about the status of our manuscript. We have revised the manuscript to address the concerns of reviewers and editorial office. The point-to-point address of all reviewers’ comments are presented below.
Hope you find revised manuscript suitable for publication in Sustainability.
We look forward to hearing from you in due course.
Yours sincerely
Dr. Duong Duc La
-----------------------------------------------------------------
Reviewer #1:
- The novelty of the manuscript is too vague and not clear, it should be clearly stated in the introduction part.
Response: Thank you very much indeed for the comment. The novelty of the work was rewritten to be clearer. Please see the yellow highlight in the manuscript.
- The mechanism for the enhanced synthesis of the Se nanoparticles by addition of the ethanol need to be clarified in the first part of discussion section.
Response: Thank you very much for the valuable comment. The mechanism for the enhanced synthesis of the Se nanoparticles by addition of the ethanol was further clarified. Please see the yellow highlight in the manuscript.
- More discussion on the mechanism of SeNPs formation by plasma solution should be added and proved
Response: Thanks for your comment. More discussion on the mechanism of SeNPs formation by plasma solution was added. Please see the yellow highlight in the manuscript.
- Author should include receive review articles (Chem.Soc.Rev. and Chem. Rev.) by Bhosale et al. to expand the work
Response: Thanks for your recommendation. The recent reviews have been added and discussed. Please see the yellow highlight in the manuscript.
- The whole manuscript should be edited by a native English speaker or an editing service
Response: Thank you for the comment. The whole manuscript was carefully checked and revised by a native English speaker. Please see the yellow highlight in the manuscript.
- Please carefully check the format and references style of the manuscript followed the author’s guide from the Sustainability journal.
Response: The format and references style of the manuscript was carefully checked.
Reviewer 2 Report
The Se NPs were successfully fabricated using the solution plasma approach, which is considered as a green synthesis. The acute oral toxicity of the prepared Se nanoparticles toward mice was successfully studied and discussed. The work presents some reasonable results, which could be suitable for publication in the Sustainability journal. However, many errors were still contained in the manuscript, which requires the author to deliver a major revision before it should be reconsidered for publication. The specific points are indicated as followings: 1. More keywords should be added to illustrate the specific fields of the research. 2. The novelty of the work should be clearly stated in the introduction part. 3. Origin and specification of several chemicals used for acute study should be provided in the material section. 4. The details of the Energy dispersive instruments to obtain the EDX results must be provided in the experimental section. 5. Details of the Raman spectroscopy should also be provided in the experimental section. 6. The author claimed that the formation of the organic compounds during the plasma process in ethanol/water solution, these newly formed organic compounds could protect the Se NPs from oxidation. More probes should be carried out to demonstrate this claim (such as TOC analysis of the solution before and after the solution plasma process should be implemented). 7. What is the main role of ethanol in the mixture for the plasma process? 8. The word format in Figure 6 should be enlarged for better vision. 9. There still exists English grammar and errors throughout the manuscript. The manuscript should be carefully checked and revised by a native English speaker
Author Response
August 15, 2022
Sustainability
Re: Response on Manuscript ID 1852983:
Thank you for considering our manuscript for publication in Sustainability. We are very grateful for such a good news about the status of our manuscript. We have revised the manuscript to address the concerns of reviewers and editorial office. The point-to-point address of all reviewers’ comments are presented below.
Hope you find revised manuscript suitable for publication in Sustainability.
We look forward to hearing from you in due course.
Yours sincerely
Dr. Duong Duc La
-----------------------------------------------------------------
- More keywords should be added to illustrate the specific fields of the research.
Response: Thank you very much for the valuable comment. The more keywords were added. Please see the yellow highlight in the manuscript.
- The novelty of the work should be clearly stated in the introduction part.
Response: Thank you very much indeed for the comment. The novelty of the work was rewritten to be clearer. Please see the yellow highlight in the manuscript.
- Origin and specification of several chemicals used for acute study should be provided in the material section.
Response: Thanks for your comment. The chemicals employed for the acute study was added. Please see the yellow highlight in the manuscript.
- The details of the Energy dispersive instruments to obtain the EDX results must be provided in the experimental section.
Response: Thank you very much for the valuable comment. The information about the EDX instrument was provided. Please see the yellow highlight in the manuscript.
- Details of the Raman spectroscopy should also be provided in the experimental section.
Response: Thank you very much for the valuable comment. The information about the Raman instrument was provided. Please see the yellow highlight in the manuscript.
- The author claimed that the formation of the organic compounds during the plasma process in ethanol/water solution, these newly formed organic compounds could protect the Se NPs from oxidation. More probes should be carried out to demonstrate this claim (such as TOC analysis of the solution before and after the solution plasma process should be implemented).
Response: Thank you very much indeed for the valuable comment. In order to confirm the presence of organic compounds on the surface of the Se NPs as probes of forming organic compounds in the solution after the plasma process, we performed the FTIR analysis. The FTIR figure and discussion were added. Please see the yellow highlight in the manuscript. Since the FTIR study is adequate to confirm the presence of protective organic coatings on the surface of Se NPs, thus in this work, we don’t perform TOC analysis.
- What is the main role of ethanol in the mixture for the plasma process?
Response: Thank you for the question. It has been demonstrated that by adding ethanol in the reaction media, the plasma-mediated synthesis of nanoparticles could significantly accelerate. The presence of the ethanol in the reaction solution also improve the hydrogen bond’s stability in the network between water molecules; as a result, changing the surface properties of the reaction media. Please see the yellow highlight in the manuscript.
- The word format in Figure 6 should be enlarged for better vision.
Response: Thank you very much for the recommendation. The word format in Figure 6 was revised for better vision. Please see the revised Figure 6 in the manuscript.
- There still exists English grammar and errors throughout the manuscript. The manuscript should be carefully checked and revised by a native English speaker.
Response: Thank you for the comment. The whole manuscript was carefully checked and revised by a native English speaker. Please see the yellow highlight in the manuscript.